# Integrating Proteomics and Metabolomics Approaches to Elucidate the Mechanism of Responses to Combined Stress in the Bell Pepper (*Capsicum annuum*)

**DOI:** 10.3390/plants13131861

**Published:** 2024-07-05

**Authors:** Brandon Estefano Morales-Merida, Jesús Christian Grimaldi-Olivas, Abraham Cruz-Mendívil, Claudia Villicaña, José Benigno Valdez-Torres, J. Basilio Heredia, Rubén Gerardo León-Chan, Luis Alberto Lightbourn-Rojas, Juan L. Monribot-Villanueva, José A. Guerrero-Analco, Eliel Ruiz-May, Josefina León-Félix

**Affiliations:** 1Laboratorio de Biología Molecular y Genómica Funcional, Centro de Investigación en Alimentación y Desarrollo, A.C., Carretera a Eldorado Km 5.5, Campo el Diez, Culiacán 80110, Sinaloa, Mexico; 2CONAHCYT—Instituto Politécnico Nacional, Centro Interdisciplinario de Investigación para el Desarrollo Integral Regional Unidad Sinaloa, Guasave 81101, Sinaloa, Mexico; 3CONAHCYT—Laboratorio de Biología Molecular y Genómica Funcional, Centro de Investigación en Alimentación y Desarrollo, A.C., Carretera a Eldorado Km 5.5, Campo el Diez, Culiacán 80110, Sinaloa, Mexico; 4Laboratorio de Genética, Instituto de Investigación Lightbourn, A.C., Carretera las Pampas Km 2.5, Jiménez 33980, Chihuahua, Mexico; 5Red de Estudios Moleculares Avanzados, Instituto de Ecología, A.C., Carretera Antigua a Coatepec 351, Congregación el Haya, Xalapa 91073, Veracruz, Mexico

**Keywords:** *Capsicum annuum*, cold, UV-B radiation, proteome, metabolome, integrating omics

## Abstract

Bell pepper plants are sensitive to environmental changes and are significantly affected by abiotic factors such as UV-B radiation and cold, which reduce their yield and production. Various approaches, including omics data integration, have been employed to understand the mechanisms by which this crop copes with abiotic stress. This study aimed to find metabolic changes in bell pepper stems caused by UV-B radiation and cold by integrating omic data. Proteome and metabolome profiles were generated using liquid chromatography coupled with mass spectrometry, and data integration was performed in the plant metabolic pathway database. The combined stress of UV-B and cold induced the accumulation of proteins related to photosynthesis, mitochondrial electron transport, and a response to a stimulus. Further, the production of flavonoids and their glycosides, as well as affecting carbon metabolism, tetrapyrrole, and scopolamine pathways, were identified. We have made the first metabolic regulatory network map showing how bell pepper stems respond to cold and UV-B stress. We did this by looking at changes in proteins and metabolites that help with respiration, photosynthesis, and the buildup of photoprotective and antioxidant compounds.

## 1. Introduction

Climate change has caused an increase in the severity and incidence of abiotic stress situations such as heat waves, drought, flooding, salinity, UV-B radiation, and cold. Plant productivity is negatively affected by a changing global climate [1]. When plants are under abiotic stress, many factors are regulated at the physiological, biochemical, and molecular levels [2]. Different molecular levels of regulatory networks control how plants respond to stress. These levels include sensing, signal transduction, transcription, transcript processing, translation, post-translational protein modifications, and changes in metabolite profiles [3,4]. These alterations can be analyzed through systems biology approaches, including genomics, transcriptomics, proteomics, and metabolomics [5,6]. Proteomics studies plant proteomes and protein functions, focusing on their roles as structural and regulatory proteins in several processes, such as growth, development, and coping against different environmental stresses in plants [7,8]. At the same time, metabolomics is a powerful tool for identifying and examining plant metabolites, and this approach can provide novel insights into how plants respond to abiotic stress [9,10]. The data produced by these systems can be enormous, sometimes without apparent interconnections. This has led to the proposal of integrating omic data to identify novel gene targets with differential expression or involvement in key metabolic pathways [11,12]. 

Bell pepper (*Capsicum annuum* L.) is an important plant in the *Solanaceae* family due to its economic, medicinal, nutraceutical, and food value [13,14]. Moreover, *C. annuum* is affected by biotic and abiotic stresses [15,16]. Recent research has reported using integrative omic approaches in *C. annuum* fruit development [17,18,19,20] and responses to abiotic stress such as cold, salinity, heat, and high-intensity light [21,22,23,24,25,26]. In addition, both cold and UV-B radiation can have an impact on plant morphology. This combined stress leads to a decrease in the germination rate of seeds, as well as reductions in plant height, leaf area, root diameter, root length, and the number of root tips in cotton seedlings [27]. *Arabidopsis* also experiences similar effects, such as reduced biomass, rosette diameter, and leaf area [28].

Regarding physiological and biochemical changes, it has been reported that combined UV-B and cold stress decreases the content of chlorophyll a and b and increases the content of chlorogenic acid, total flavonoids, and carotenoids in bell pepper leaves [29]. Furthermore, UV-B and cold stress significantly increase the accumulation of phenolic compounds like kaempferol, hydroxycinnamic acids, and quercetin derivatives in *Arabidopsis*, as well as apigenin 7-O-glucoside (A7G) and luteolin 7-O-glucoside (L7G) in bell pepper, which serve as antioxidants and absorb UV-B radiation [29,30,31]. 

Researchers have also found changes in gene expression in the stem of *C. annuum*, particularly in some genes (*ANS*, *DFR*, *F3H*, *F3′5’H*, and *MYB*) involved in flavonoid biosynthesis, such as flavonols and anthocyanins [32]. Also, when UV-B and cold stress were combined in *Arabidopsis,* cold-responsive genes were induced, such as *CBF1*, *CBF2*, *CBF3*, *COR15A*, *COR15B*, and *COR78*, as well as flavonoid biosynthesis genes like *MYB111*, *CHI*, and *FLS* [31]. In a previous study, we discovered the down-regulation of several genes involved in cell wall growth and pathogen defense. Conversely, the up-regulated genes were associated with protecting the chloroplast, transporting compounds, and synthesizing hormones and flavonoids [33].

Research on plant responses to stress combinations is essential, as it is impossible to accurately predict these responses by only examining how plants react to individual stressors [34,35,36,37]. In such a way, stress-specific proteome and metabolome integration can identify unique mechanisms that respond to combined stress from UV-B radiation and cold. However, at present, there is not sufficient information about changes in the proteome and metabolome profiles of bell peppers under combined UV-B radiation and cold stress.

Therefore, the purpose of this study was to find metabolic changes in bell pepper stems caused by UV-B radiation and cold by integrating omic data.

## 2. Results

### 2.1. Differentially Expressed Proteins (DEPs) Identification

To investigate the functions of proteins in *C. annuum* under combined cold stress and UV-B, we collected stress-treated and control stem proteins and used an LC MS/MS technique to create proteome profiles.

Control and UV-B+cold samples exhibited similar protein profiles on an SDS-PAGE (Appendix A). Principal component analysis (PCA) was used to assess the relationship between biological replicates, allowing us to affirm that the data are reliable. These data indicate that 70.6% of variation corresponds to a regulation of protein abundance due to combined stress (Figure 1A). In the global proteome, a total of 1134 proteins were discovered. Based on log_2,_ folds were considered of more than 0.50 (over-accumulated) and less than −0.50 (down-accumulated) at a *p*-value < 0.05. A total of 200 DEPs were identified (Appendix A), of which 129 were over-accumulated and 71 were down-accumulated (Figure 1B).

Furthermore, GO enrichment analysis and the clustering of redundant terms were conducted to assess the functions of DEPs (Appendix A). In the combined UV-B+cold samples, we found over-accumulation of proteins related to the control of specific biological processes such as carbon fixation, developmental process, generation of precursor metabolites and energy, glyoxysome organization, multicellular organismal process, regulation of DNA methylation, response to stimulus, seedling development, and tricarboxylic acid transmembrane transport (Figure 1C). This investigation also showed a negative regulation of the biosynthetic process and heterochromatin organization (Figure 1D). 

### 2.2. Untargeted Metabolomic Analyses

In both negative and positive ionization modes, untargeted metabolomic analyses were carried out on stems from the control and the UV-B+cold groups. The PCA accounted for 95.8% of the total variation, and it shows that the metabolomic profiles of the control and the UV-B+cold plants did not overlap, meaning they have different phytochemical signatures (Figure 2A). We found only 37 over-accumulated (Log_2_ FC: ≥2) metabolites (Appendix A). The normalized abundance of metabolites in each sample was plotted in a heatmap, where two main blocks were observed. The first block includes the control plants, and the second consists of the UV-B+Cold stress (Appendix A). 

The findings of the pathway analysis are depicted in Figure 2B. We identified 29 pathways, with only 7 showing significant enrichment (Appendix A). The main pathways were flavone and flavonol biosynthesis (quercitrin [LFC: 3.94], quercetin 3-O-rhamnoside 7-O-glucoside [LFC: 2.97], luteolin [LFC: 2.94], and kaempferol 3-O-rhamnoside-7-O-glucoside [LFC: 2.94]), flavonoid biosynthesis (eriodictyol [LFC: 3.04], luteolin [LFC: 2.94], chlorogenic acid [LFC: 2.92], dihydrokaempferol [LFC: 2.85], kaempferol [LFC: 2.74], and delphinidin [LFC: 2.67]), tropane, piperidine and pyridine alkaloid biosynthesis (L-phenylalanine [LFC: 3.10], tropine [LFC: 2.88], and tropinone [LFC: 2.75]), phenylpropanoid biosynthesis (L-phenylalanine [LFC: 3.10], *p*-hydroxyphenyl lignin [LFC: 3.07], chlorogenic acid [LFC: 2.92], and 5-hydroxyconiferaldehyde [LFC: 2.91]), galactose metabolism (sucrose [LFC: 5.08], D- glucose [LFC: 2.91], and raffinose [LFC: 2.59]), phenylalanine metabolism (L-phenylalanine [LFC: 3.10] and phenylacetic acid [LFC: 3.03]), and tyrosine metabolism (4-hydroxyphenylacetaldehyde [LFC: 3.23] and 4-hydroxyphenylpyruvic acid [LFC: 2.89]).

### 2.3. Targeted Metabolomic Analyses

According to the untargeted metabolomic analyses, we performed specific assays that examined the phenolic content to verify the previously mentioned results. In addition to the amino acid phenylalanine, 15 compounds from various chemical sub-categories were discovered and measured in steam samples (Table 1). In the control samples, the concentrations of compounds such as phenylalanine, 4-hydroxybenzoic acid, vanillic acid, 4-coumaric acid, 3-coumaric acid, ferulic acid, salicylic acid, vanillin, luteolin-7-O-glucoside, rutin, and penta-O-galloyl-B-D-glucose were higher than those in the UV-B+cold samples. The control and UV-B+cold samples had about the same amount of protocatechuic acid and quercetin-3-glucoside. Finally, we found higher amounts of the phenolics chlorogenic acid, luteolin, and quercitrin in the UV-B+cold samples.

### 2.4. Data Integration

The analysis of data integration using proteomics and untargeted and targeted metabolomics showed alterations in the accumulation of proteins and metabolites associated with flavonoid biosynthesis, the superpathway of glycolysis and the TCA cycle, porphyrin metabolism, and scopolamine biosynthesis (Figure 3). 

In the flavonoid biosynthesis pathway, flavonoid aglycones and their glycosylated derivatives such as eriodictyol, dihydrokaempferol, kaempferol, kaempferol 3-O-α-L-rhamnoside, kaempferol 3-O-rhamnoside-7-O-glucoside, luteolin, apiin (apigenin 7-O-[β-D-apiosyl-(1→2)-β-D-glucoside]), quercitrin (quercetin 3-O-rhamnoside), and quercetin 3-O-rhamnoside 7-O-glucoside displayed positive accumulation (stress/control). In contrast, luteolin-7-O-glucoside, quercetin-3-glucoside, and rutin were down-accumulated (stress/control).

Regarding the superpathway of glycolysis and TCA cycle, enzymes like fructose-bisphosphate aldolase (FbaA; PHT93527 and PHT77434), glyceraldehyde-3-phosphate dehydrogenase (GAPDH; PHT76144, PHT66665, and PHT89152), succinyl-CoA synthetase (SCS; PHT89296), and malate dehydrogenase (MDH; PHT95662) showed changes in accumulation. Stress-treated samples showed a significant increase in malate and glutamic acid levels.

Heme b biosynthesis is the best-known pathway precursor for tetrapyrroles (chlorophyll a and b). We detected over-accumulation of precursor molecules, including glutamic acid, protoporphyrin IX, and biliverdin IXα, as well as enzymes such as glutamate-1-semialdehyde aminotransferase (PHT81091) and coproporphyrinogen III oxidase (PHT70068).

Finally, scopolamine biosynthesis was observed, where tropinone, tropine, and one enzyme, tropinone reductase (PHT71246), were over-accumulated.

## 3. Discussion

The current work is the first attempt to integrate proteomic and metabolomic approaches to enhance our comprehension of the mechanisms involved in bell pepper plants’ responses when exposed to combined UV-B and cold stress.

### 3.1. Combined UV-B and Cold Stress Induces Changes in Flavonoid Biosynthesis

Integrative analysis revealed changes in the accumulation of metabolites related to flavonoid biosynthesis. It is well known that UV-B rays damage DNA, proteins, lipids, and chloroplast membranes, as well as various elements of photosystem II [38,39]. Due to their varying UV absorption capabilities, flavonoids can serve as a crucial defense mechanism for plants, acting as a protective barrier when exposed to UV radiation [40]. Similarly, flavonoids, including flavonols, flavanols, flavones, and anthocyanins, play a crucial role in determining resistance to freezing and adaptation to cold. The absence or reduction in these compounds reduces antioxidant and reactive oxygen species scavenger activity, causing harm to the system responsible for freezing resistance [41,42]. Previous research has identified an improved content of flavonoids in *Arabidopsis* when exposed to a combination of UV radiation and cold stress [31]. In bell peppers, there has been an increase in gene expression of the *ANS*, *CHI*, *CHS*, *DFR*, *F3H*, *F3′5′H*, *FLS*, and *MYB* genes, as well as an increased accumulation of L7G and A7G when plants were exposed to low temperatures and UV-B radiation [29,32,33]. Notably, our research reveals that five of the nine metabolites accumulated in flavonoid biosynthesis are flavonoid glycosides. According to earlier studies, treating the plants with cold and UV-B stress increased the amount of flavonoid glycosides [43,44,45,46,47,48]. These alterations of flavonoids enhance their structural stability, solubility at the molecular level, and capacity to be transported into vacuoles and chloroplasts during storage [49,50]. Interestingly, this study also identified glucosides (glucose derivatives) in four of the five flavonoid glycosides, suggesting a connection to the accumulation of sugars such as D-glucose, raffinose, and sucrose, as observed here. These observations highlight the important role that flavonoids have as photoprotectors and antioxidants in the response to combined stress, with an emphasis on flavonoid glycosides.

### 3.2. Effects of Combined UV-B and Cold Stress on Respiration and Photosynthesis Metabolism

In our data, glycolysis and TCA cycle changes were found. Cold stress in *C. annum* altered proteins related to glycolysis and the TCA cycle, such as GAPDH, FbaA, and MDH, at 2 h and 12 h after exposure, and after 1 h recovery after 72 h of exposure [51]. This investigation also identified these proteins. GAPDHs are important for many cellular functions in plants, such as energy generation, DNA repair, gene expression control, sugar and amino acid levels maintenance, and the transmission of signals related to abscisic acid (ABA) [52,53,54]. Additionally, the expression of GAPDHs in the shoots and roots of *Arabidopsis* and *Triticum aestivum* was induced by drought, salt, osmotic stress, heat, and cold stress [55,56]. On the other hand, FbaA is an enzyme crucial for plant carbon metabolism, including glycolysis, gluconeogenesis, and photosynthesis [57]. A study in potatoes demonstrated the importance of FbaA. Transgenic potato plants with reduced levels of the *FbaA* gene demonstrate reductions in photosynthesis, carbon metabolism, and growth [58]. Also, cold, heat, drought, as well as blue and red light modify the gene expression of the FBA gene family in *Solanaceae*, such as *Nicotiana tabacum*, *Solanum tuberosum*, and *Solanum lycopersicum* [59,60,61,62]. Meanwhile, MDH and its catalytic product malate (also identified in this study) play a role in Arabidopsis respiration, CO_2_ release in the photorespiratory pathway, fatty acid β-oxidation, redox homeostasis, P uptake, and fixing N [63,64,65,66,67]. In *S. tuberosum* and *S. lycopersicum*, MDH gene family were shown to participate in plant responses to several abiotic stressors, such as cold, heat, drought, and salt [68,69]. Additionally, the proteomic data revealed that stress samples over-accumulated two cytochrome c oxidase enzymes (PHT79443 and PHT66082), ATP synthase subunit α (PHT94639), and ATP synthase subunit F (PHT89781), which are involved in mitochondrial electron transport from cytochrome c to oxygen. These data suggest that respiratory metabolism may play a role in response to combined UV-B and cold stress.

Heme b biosynthesis is part of tetrapyrroles, essential for nitrate and sulfate assimilation, the detoxification of reactive oxygen species, respiration, photosynthesis, and light signaling [70]. Previous studies have reported that UV-B radiation and cold affect photosynthesis individually. UV-B radiation causes direct and indirect harm to photosystem II and photosystem I, decreasing chlorophyll levels [71,72,73]. Meanwhile, cold generates oxidative damage in chloroplasts and the photoinhibition of PSII, affecting electron transport and chlorophyll content [74,75,76]. Furthermore, analysis of the supercluster GO data revealed the positive regulation of photosynthesis-related proteins (inside of the generation of precursor metabolites and energy), including PSAA (PHT68443), PSBC (PHT78035), MPH2 (PHT73773), ELIP (PHT85861), and CYP38 (PHT85861) (Figure 1C). We detected the over-accumulation of glutamic acid (Glu), protoporphyrin IX (Proto-IX), and biliverdin IXα. Glu is the first precursor to tetrapyrroles, and its levels increase under cold and salinity stress in *A. thaliana* [77,78]. Proto-IX is the final shared precursor for chlorophyll and heme synthesis, and for metabolites related to respiration and photosynthesis [77]. Some investigations have demonstrated the importance of Proto-IX in plant development. Herbicides such as Pyraflufen ethyl and Saflufenacil inhibit Proto IX accumulation, causing plant death [79,80,81]. Further, researchers have looked into how different abiotic factors, like iron deficiency, salinity, drought, heat, and cold, affect the production of porphyrins like Proto-IX, Mg-Proto-IX and its methyl ester, and protochlorophyllide [82,83,84,85]. Regarding biliverdin IXα, it is known as a precursor to synthesizing phytochromes, sensory photoreceptors that modulate plant growth [86]. These findings suggest that metabolites, intermediates of chlorophyll and heme synthesis, and proteins involved in photosynthesis and their associated complex regulatory networks, contribute to the response to combined stress.

### 3.3. Combined UV-B and Cold Stress Induces Changes in Tropine Biosynthesis

In this study, tropinone, tropine, and tropinone reductase (TRI; PHT71246) contents showed an increase in combined UV-B and cold stress samples. Tropinone and tropine are tropane alkaloids that exhibit effects in humans as antidepressants, spasmolytics, local anesthetics, improve blood flow in the microcirculation, and reduce asthmatic symptoms [87]. Researchers have discovered that Solanaceae plants use tropane alkaloids to defend against insects, predators, and pathogens [88]. Recent reports indicate that *C. annuum* possesses the enzymes necessary for producing tropinone and tropine [89]. This demonstrates that UV-B radiation and cold are causing the accumulation of these compounds. However, the role of these compounds in plants under abiotic stress is still unknown.

## 4. Materials and Methods

### 4.1. The Plant Materials

Bell pepper seeds, cultivar cannon (Zeraim Gedera-Syngenta, Gedera, Israel), were sprouted and maintained as previously mentioned [29]. Comprehensive information may be found in Section A.1.

Seedlings were placed in a plant growth chamber (GC-300TLH, JEIO TECH; Seoul, Republic of Korea) 28 days after sowing for three days (from day 28 to 30). The conditions were as follows: temperature 25/20 °C (day/night), relative humidity 65%, and photoperiod of 12 h (from 6:00 to 18:00 h) of photosynthetically active radiation (PAR) (972 μmol m^−2^ s^−1^). After three days, the control plants were kept under the previously described conditions. For the UV-B+cold stress treatment, the conditions were as follows: temperature 15/10 °C (adjusted on day 30 at 18:00), photoperiod of PAR for 6 h (from 06:00 to 10:00 and from 16:00 to 18:00 h), and UV-B radiation (72 kJ m^2^) for 6 h (from 10:00 to 16:00 h) for 2 days (days 31 and 32). The UV-B radiation was applied in accordance with the methodology outlined by León-Chan et al. [29]. Control and UV-B+cold samples were collected at 11:00 on day 32 (Appendix A). The experiment was repeated thrice, and 20 stems were sampled for each repetition, flash-frozen in liquid nitrogen, and stored at −80 °C. 

### 4.2. Proteomic Analysis

According to Monribot-Villanueva et al.’s instructions, the proteomic analysis included protein extraction, digestions, peptide fractionation, TMT labeling, SPS-MS3, and data processing [90,91]. The proteomic analysis specifications are in Section A.2.

Three biological replicates of stem powder (750 mg) were used for protein extraction with the phenol–acetone method. In accordance with the manufacturer’s directions, Tandem Mass Tag (TMT) 6-plex reagents were used in the following order: 126, 127N, and 127C for control samples, and 130N, 130C, and 131 for UV-B+cold samples. We used the *C. annuum* UniProt Reference proteome (UP000222542) as the database for protein identification. A protein was considered over-accumulated and down-accumulated if its *p*-value < 0.05 and relative Log_2_-fold change >0.50 and <−0.50 (Stress/Control), respectively. The ggplot2 (v3.5.1) package created all graphs [92].

### 4.3. Metabolomic Analyses

Methanolic extracts were subjected to untargeted and targeted metabolomic analyses, following the methodology described by Monribot-Villanueva et al. [90,91]. Comprehensive information may be found in Section A.3.

Untargeted metabolomic analyses were performed using an ultra-high-resolution chromatograph coupled to a high-resolution mass spectrometer (UPLC-HRMS-QTOF; Class I-Synapt G2-Si, Waters™, Milford, MA, USA). We employed both positive and negative ionization modes during the analyses. The metabolomic data were initially analyzed using MarkerLynx (v4.1, Waters™, Milford, MA, USA) and MassLynx (v4.1, Waters™, Milford, MA, USA) software. The MetaboAnalyst bioinformatic platform (https://www.metaboanalyst.ca, accessed on 11 October 2023) was utilized to conduct analyses on the untargeted produced data, employing its many modules [93]. The Statistical Analysis module was used to conduct fold change (stress/control) and FDR studies to identify either over-accumulated or down-accumulated features. Over-accumulated features were defined as having Log_2_ FC values ≥ 2, while down-accumulated features were described as having Log_2_ FC values ≤ −2 and FDR values ≤ 0.01 for both. The *m/z* signals were provisionally identified by utilizing the Functional Analysis module. The metabolites that exhibited excessive accumulation and reduced accumulation in both ionization modes were subsequently connected and subjected to analysis using the Pathway Analysis module.

The targeted metabolomic quantification of phenolic chemicals was conducted on a UPLC-QqQ mass spectrometer (Agilent Technologies 1290/6460; Santa Clara, CA, USA), as previously described by Juárez-Trujillo et al. and Monribot-Villanueva et al. [90,94]. Calibration curves were generated for each chemical (60 compounds) in the 0.25 to 17 µM range for quantification purposes. The coefficient of determination (r^2^) for quadratic regressions was 0.99. Agilent Technologies’ MassHunter software (vB.06.00) processed the data. Statistical analyses were conducted using the Rstudio program (v2023.09.1+494) to identify significant differences (*p*-value < 0.05) among the samples. Phenylalanine, protocatechuic acid, chlorogenic acid, 3-coumaric acid, salicylic acid, luteolin, luteolin-7-O-glucoside, quercetin-3-glucoside, quercitrin, rutin, and penta-O-galloyl-B-D-glucose exhibited normal distributions, so a *t*-test was conducted. The compounds 4-hydroxybenzoic acid, vanillic acid, 4-coumaric acid, and vanillin exhibited a non-normal distribution, requiring the Wilcoxon rank sum test. The ggplot2 (v3.5.1) package was used to create all graphs [92].

### 4.4. Integration Data

The metabolic pathways were constructed with Uniprot annotation (https://www.uniprot.org/, accessed on 14 January 2024) and plant metabolic pathway databases (https://plantcyc.org/, accessed on 21 January 2024) using *C. annuum* as a reference. The log_2_-fold change values of proteins and metabolites that showed differences were entered into cellular overview/omics viewer programs to represent the metabolic pathways visually [95]. This analysis was based on a pathway perturbation score (PPS) value of 50. The purpose of the PPS is to measure the amount of activation of a certain route at a specific moment in time. The Pathway Collage tool was used to graph the metabolic pathways activated by combined UV-B and cold stress. 

## 5. Conclusions

This study presents the first approach to identifying the key mechanisms involved in the pepper response to combined stress using proteomic and metabolomic data integration. This analysis suggests that flavonoids and their glycosides participate in absorbing UV-B light and acting as antioxidants. Carbon metabolism showed important changes in proteins and metabolites related to energy generation through cellular respiration. Porphyrin metabolism has proven to be important because it contributes to chlorophyll and heme synthesis, two metabolites involved in photosynthesis. The biosynthesis of scopolamine indicates that bell pepper plants can use combined UV-B and cold stress as an inducer to produce bioactive compounds like tropine and tropinone.

## Figures and Tables

**Figure 1 plants-13-01861-f001:**
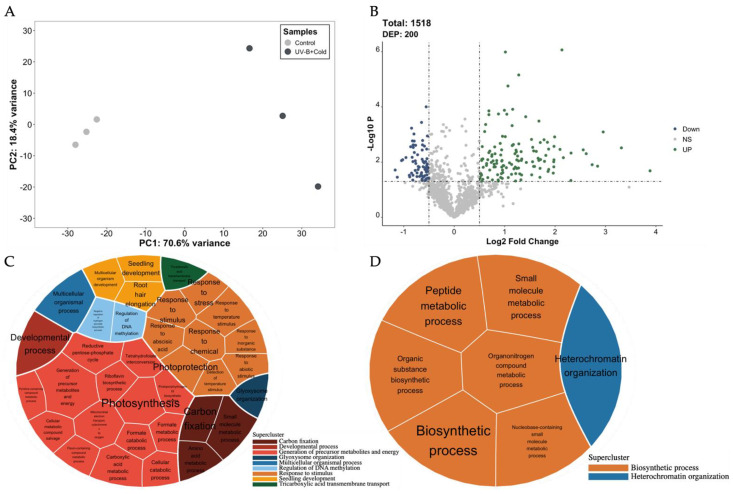
Proteomic study of bell pepper stems exposed to both UV-B and cold stress. Principal component analysis of the control (gray) and UV-B+cold (black) biological replicates (**A**). Volcano graph showing differentially accumulated proteins (**B**). Green dots indicate over-accumulated proteins, while blue dots indicate down-accumulated ones. In the *X* axis, the logarithmic fold change is represented, and in the *Y* axis, the *p*-value is represented in scale as −Log10. Voronoi plot of the Gene Ontology (GO) analysis of the differentially expressed proteins, where categories of GO were grouped into superclusters. Over-accumulated proteins (**C**). Down-accumulated proteins (**D**) based on the absolute log10 *p*-values.

**Figure 2 plants-13-01861-f002:**
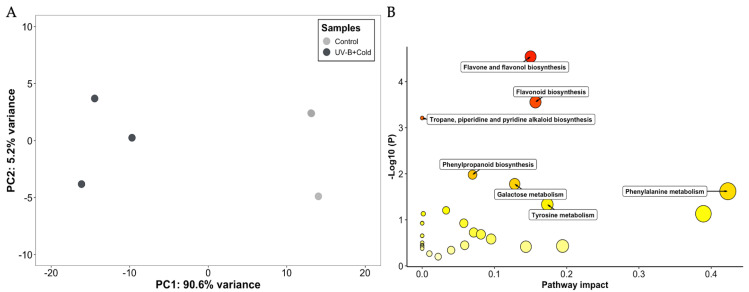
Untargeted metabolomics analysis of bell pepper stems exposed to both UV-B and cold stress. The principal component analysis was conducted on the control (gray) and UV-B+cold (black) biological replicates (**A**). The pathway analysis module of MetaboAnalyst identified metabolic pathways in response to UV-B+cold (**B**). Each circle represents a metabolic pathway: the circle’s color indicates its *p*-value; the circle’s size represents the pathway impact.

**Figure 3 plants-13-01861-f003:**
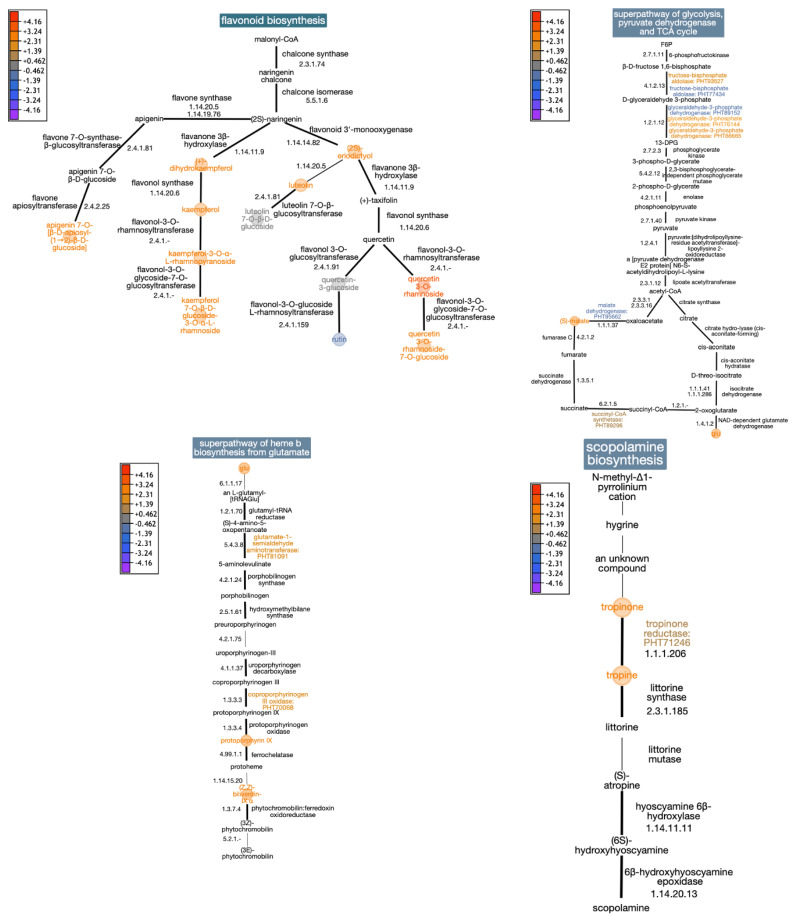
Plant metabolic pathways are based on data from flavonoid biosynthesis, carbon metabolism, heme biosynthesis, and scopolamine biosynthesis (https://plantcyc.org/, accessed on 21 January 2024). The *C. annuum* database was used. The color scale is based on the log2-fold change value (stress/control) for the relative content of metabolites and enzymes. The thick black lines indicate the purification and functional characterization of the enzymes. In comparison, thin lines indicate that enzymes have not been functionally characterized.

**Table 1 plants-13-01861-t001:** Phenolic compound profiling was performed using metabolomic analyses in stem of bell pepper under combined UV-B and cold stress.

Compound	Samples	
	Control	UV-B+Cold
Phenylalanine ^1^	101.75 ± 2.39 ^a^	91.78 ± 1.30 ^b^
Protocatechuic acid ^1^	0.31 ± 0.01 * ^a^	0.29 ± 0.03* ^a^
4-hydroxybenzoic acid ^2^	1 ± 0.05 ^a^	0.69 ± 0.00 ^a^
Vanillic acid ^2^	1.04 ± 0.01 ^a^	0.55 ± 0.03 ^a^
Chlorogenic acid ^1^	22.89 ± 0.20 ^a^	57.17 ± 0.03 ^b^
4-Coumaric acid ^2^	0.26 ± 0.01 * ^a^	0.11 ± 0.01* ^a^
3-Coumaric acid ^1^	3.61 ± 0.03 ^a^	3.04 ± 0.06 ^b^
Ferulic acid ^3^	0.05 ± 0.01 *	---
Salicylic acid ^1^	2.31 ± 0.14 ^a^	0.30 ± 0.06* ^b^
Vanillin ^2^	1.44 ± 0.06 ^a^	0.92 ± 0.02 ^a^
Luteolin ^1^	0.66 ± 0.01 * ^a^	0.72 ± 0.04* ^a^
Luteolin-7-O-glucoside ^1^	82.86 ± 2.17 ^a^	63.99 ± 3.38 ^b^
Quercetin-3-glucoside ^1^	4.97 ± 0.12 ^a^	4.93 ± 0.05 ^a^
Quercitrin ^1^	3.97 ± 0.05 ^a^	4.34 ± 0.05 ^b^
Rutin ^1^	9.17 ± 0.33 ^a^	6.31 ± 0.19 ^b^
Penta-O-galloyl-B-D-glucose ^1^	1.47 ± 0.42 * ^a^	0.32 ± 0.05* ^b^

The unit for concentration is μg/g dry matter. Values show mean ± standard deviation (n = 3). * Data that fall below the quantification threshold. --- Data beneath the detection threshold. ^1^: *t*-test analyses. ^2^: Wilcoxon rank sum test analyses. ^3^: Non-statistical analyses. Significant disparities between UV-B+cold and control samples for each compound are indicated using different letters.

## Data Availability

Data are contained within the article and Appendix A.

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
