# Peer review of "Integrating Proteomics and Metabolomics Approaches to Elucidate the Mechanism of Responses to Combined Stress in the Bell Pepper (*Capsicum annuum*)"

_plants, 2024, doi:10.3390/plants13131861_

Round 1

Reviewer 1 Report

Comments and Suggestions for Authors

The manuscript “Integrating proteomics and metabolomics approaches to elucidate the mechanism of responses to combined stress in bell pepper (Capsicum annuum)” signifies the response of Bell pepper plants against different abiotic stresses.

Summary of the study:

Bell pepper plants are sensitive to environmental changes like UV-B radiation and cold, impacting yield. Omic data integration is used to understand how plants cope with abiotic stress, providing insights into metabolic responses. This study aimed to identify metabolic changes in bell pepper stems under combined UV-B radiation and cold stress. Proteome and metabolome data were generated using liquid chromatography coupled with mass spectrometry. Integration of protein and metabolite profiles revealed induction of flavonoids, carbon metabolism, tetrapyrrole, and scopolamine pathways under combined stress. The study highlights mechanisms involved in the response to UV-B and cold stress, including flavonoid biosynthesis, respiration, and photosynthesis pathways.

This MS needs significant improvements, specifically due to high plagiarism which is 36%.

Reviewer comments:

1-      Why plagiarism is more than 36%? Please rewrite the plagiarized sentences.

2-      In abstract, line 23-31: “the authors described so much back ground and methods of this study, I think 2-3 line for background, 1-2 line for methods/approach, and 6-8 lines for results and then conclusion. Please rewrite the abstract, and emphasize on the keys results of this study in the abstract section.

3-      Figure 2A: why the y-axis has only 5% variance? Whereas the PC1 has more than 90%. Please recheck the results and data.

4-      Please conduct the Hierarchical cluster analysis (HCA)/heat-map analysis of metabolic results.

5-      Make a comprehensive table that shows the retention time, m/z value, FDR, and different attributes of metablomics….

6-      Line 152-153: “Different letters 152 indicate significant differences among UV-B+cold and control samples for each compound.”  Which statistical tool was applied on this data.???

Comments on the Quality of English Language

This MS needs significant improvements, specifically due to high plagiarism which is 36%.

1-      Why plagiarism is more than 36%? Please rewrite the plagiarized sentences.

Author Response

For research article

Response to Reviewer 1 Comments

First of all, thanks for the precious comments. We have modified the manuscript for review (highlighted in yellow). Below are our responses to the comments (in italics).

Summary of the study:

Bell pepper plants are sensitive to environmental changes like UV-B radiation and cold, impacting yield. Omic data integration is used to understand how plants cope with abiotic stress, providing insights into metabolic responses. This study aimed to identify metabolic changes in bell pepper stems under combined UV-B radiation and cold stress. Proteome and metabolome data were generated using liquid chromatography coupled with mass spectrometry. Integration of protein and metabolite profiles revealed induction of flavonoids, carbon metabolism, tetrapyrrole, and scopolamine pathways under combined stress. The study highlights mechanisms involved in the response to UV-B and cold stress, including flavonoid biosynthesis, respiration, and photosynthesis pathways.

This MS needs significant improvements, specifically due to high plagiarism which is 36%.

Reviewer comments:

Comments 1: Why plagiarism is more than 36%? Please rewrite the plagiarized sentences.

Response 1: Dear reviewer, thank you for your comment. We reviewed the manuscript and made a formatting error when placing the appendix within the main text; due to this, the percentage of overlap was very high. We solved this and rephrased the sentences with a high degree of similarity. In the end, we only detected several similarities between compound names, gene ontology categories, and metabolic pathways.

Comments 2: In abstract, line 23-31: “the authors described so much back ground and methods of this study, I think 2-3 line for background, 1-2 line for methods/approach, and 6-8 lines for results and then conclusion. Please rewrite the abstract, and emphasize on the keys results of this study in the abstract section.

Response 2: Dear reviewer, thank you for your comment. The paragraph was revised and corrected.

Comments 3: Figure 2A: why the y-axis has only 5% variance? Whereas the PC1 has more than 90%. Please recheck the results and data.

Response 3: We appreciate your comment. We reanalyzed the data and found the same results, which we have attached here.

Sample

PC1

PC2

PC3

PC4

PC5

PC6

UV-B+Cold_1

-14.459

3.6897

-2.8607

-1.4913

0.0070403

1.04E-14

UV-B+Cold_2

-9.7058

0.24504

-1.3904

2.5865

-0.012235

5.98E-15

UV-B+Cold_3

-16.135

-3.8226

3.6807

-0.62781

0.0019226

1.11E-16

Control_1

13.099

2.4082

1.9211

0.052441

0.574

-8.24E-15

Control_2

13.158

2.3761

1.8873

-0.027896

-0.57649

-2.91E-15

Control_3

14.043

-4.8965

-3.238

-0.49194

0.0057552

-4.49E-15

Total variance (%)

Variance (%)

90.5537638

5.2182632

3.39120472

0.78253943

0.05422882

1.9686E-29

100

Comments 4: Please conduct the Hierarchical cluster analysis (HCA)/heat-map analysis of metabolic results.

Response 4: We are grateful for the suggestion. We realized the heat-map analysis and added it to the supplementary materials (Figure S3).

Comments 5: Make a comprehensive table that shows the retention time, m/z value, FDR, and different attributes of metablomics….

Response 5: We appreciate your comment. The table you mentioned is part of the supplementary materials (Table S3); we only added the FDR column.

Comments 6: Line 152-153: “Different letters 152 indicate significant differences among UV-B+cold and control samples for each compound.”  Which statistical tool was applied on this data.???

Response 6: Dear reviewer, thank you for your comment. We used the t test and Wilcoxon test to compare the means for data with a normal and non-normal distribution, respectively. We added these tools to the table footer and described them in the methodology (lines 341–343).

Comments on the Quality of English Language

This MS needs significant improvements, specifically due to high plagiarism which is 36%.

Comments 1: Why plagiarism is more than 36%? Please rewrite the plagiarized sentences.

Response 1: Dear reviewer, thank you for your comment. We reviewed the manuscript and made a formatting error when placing the appendix within the main text; due to this, the percentage of overlap was very high. We solved this and rephrased the sentences with a high degree of similarity. In addition, the manuscript was reviewed by the colleague, Dr. Basilio Heredia, to improve writing in English. Dr. Basilio completed his doctorate at Texas A&M University. He also has a variety of English publications and has served as an editor for several journals.

Reviewer 2 Report

Comments and Suggestions for Authors

The study deals with the bell pepper response to combined cold and UV abiotic stress. The paper is interesting due to the combination of proteomic and metabolomic methods. The detailed comments are listed below:

Abstract:

L32-35: add % changes of examined parameters between stressed and non-stresses plants

Introduction:

The Introduction is very general. Description of some of the examined compounds is needed. Emphasize the ubiquitous role of antioxidant enzymes in mitigating different abiotic stresses, e.g. cold, drought, pesticides (refer to https://doi.org/10.1016/j.chemosphere.2022.136284)

Shortly describe physiological, biochemical, and molecular plant response to cold and UV stress

L65-66: indicate their role in stress mitigation

Results:

L145: what the Authors mean by higher and lower P value? Indicate P >0.05 or P <0.05

Materials and Methods:

How were plants grown?

Indicate pot dimensions, how many plants per pot?

How plants were water?

Was fertilization applied?

Composition of the substrate in the pot.

Comments on the Quality of English Language

Moderate editing of English language required

Author Response

For research article

Response to Reviewer 2 Comments

First of all, thanks for the precious comments. We have modified the manuscript for review (highlighted in green). Below are our responses to the comments (in italics).

The study deals with the bell pepper response to combined cold and UV abiotic stress. The paper is interesting due to the combination of proteomic and metabolomic methods. The detailed comments are listed below:

Abstract:

Comments 1: L32-35: add % changes of examined parameters between stressed and non-stresses plants.

Response 1: Dear reviewer, thank you for your comment. After examining your comment, we cannot place % changes since the methodology used only indicates the metabolic pathways where changes in the accumulation of proteins and metabolites occur.

Introduction: 

Comments 2: The Introduction is very general. Description of some of the examined compounds is needed.

Response 2: Thank you for pointing this out. We agree with your comment. We have, accordingly, revised and modified the introduction to emphasize this point.

Comments 3: Emphasize the ubiquitous role of antioxidant enzymes in mitigating different abiotic stresses, e.g. cold, drought, pesticides (refer to https://doi.org/10.1016/j.chemosphere.2022.136284).

Response 3: We appreciate your comment.After reviewing the topic, we conclude that a description of antioxidant enzymes is not necessary, as our study did not identify them.

Comments 4: Shortly describe physiological, biochemical, and molecular plant response to cold and UV stress.

Response 4: Thank you for pointing this out. We agree with this comment. Therefore, we add information in lines 60–76.

Comments 5:L65-66: indicate their role in stress mitigation

Response 5: Agree. We have, accordingly, modified lines 69–71 to emphasize this point.

Results:

Comment 6: L145: what the Authors mean by higher and lower P value? Indicate P >0.05 or P <0.05

Response 6: Thank you for your comment. The sentence was modified. The main point is to demonstrate how colors represent the P-Value of each pathway.

Materials and Methods:

Comment 7: How were plants grown? Indicate pot dimensions, how many plants per pot? How plants were water? Was fertilization applied? Composition of the substrate in the pot.

Response 7: We appreciate your comment. The process by which the pepper seedlings were produced may be found in Appendix A1. This paper represents a continuation of several prior investigations. This means that adding this information to the main text will result in overlaps within the manuscript.

Comments on the Quality of English Language

Comment 1: Moderate editing of English language required

Response 1: Dear reviewer, thank you for your comment. The manuscript was reviewed by the colleague, Dr. Basilio Heredia, to improve writing in English. Dr. Basilio completed his doctorate at Texas A&M University. He also has a variety of English publications and has served as an editor for several journals.

Round 2

Reviewer 1 Report

Comments and Suggestions for Authors

the authors did sufficient revisions.

Comments on the Quality of English Language

English is fine.

Reviewer 2 Report

Comments and Suggestions for Authors

The Authors have corrected the manuscript. I have no more comments